# EPIAMNE: A New Scoring System for Differentiating Transient EPIleptic AMNEsia from Transient Global Amnesia

**DOI:** 10.3390/brainsci12121632

**Published:** 2022-11-29

**Authors:** Biagio Maria Sancetta, Lorenzo Ricci, Giovanni Assenza, Marilisa Boscarino, Flavia Narducci, Carlo Vico, Vincenzo Di Lazzaro, Mario Tombini

**Affiliations:** Neurology, Neurophysiology and Neurobiology Unit, Department of Medicine, University Campus Bio-Medico of Rome, 00128 Rome, Italy

**Keywords:** acute amnestic syndromes, transient epileptic amnesia, quantitative EEG analysis

## Abstract

Transient epileptic amnesia (TEA) is a rare cause of acute amnestic syndromes (AAS), often misdiagnosed as transient global amnesia (TGA). We proposed a scoring system—the EPIlepsy AMNEsia (EPIAMNE) score—using quantitative EEG (qEEG) analysis to obtain a tool for differentiating TEA from TGA. We retrospectively reviewed clinical information and standard EEGs (stEEG) of 19 patients with TEA and 21 with TGA. We computed and compared Power Spectral Density, demonstrating an increased relative theta power in TGA. We subsequently incorporated qEEG features in EPIAMNE score, together with clinical and stEEG features. ROC curve models and pairwise ROC curve comparison were used to evaluate and compare the diagnostic accuracy for TEA detection of EPIAMNE score, presence of symptoms atypical for TGA (pSymAT) and identification of anomalies (interictal epileptiform or temporal focal spiky transients) at stEEG (PosEEG). Area Under the Curve (AUC) of EPIAMNE score revealed to be higher than PosEEG and pSymAT (AUC_EPIAMNE_ = 0.95, AUC_pSymAT_ = 0.85, AUC_PosEEG_ = 0.67) and this superiority proved to be statistically significant (*p*-value_EPIAMNE-PosEEG_ and *p*-value_EPIAMNE-pSymAT_ < 0.05). In conclusion, EPIAMNE score classified TEA with higher accuracy than PosEEG and pSymAT. This approach could become a promising tool for the differential diagnosis of AAS, especially for early TEA detection.

## 1. Introduction

Acute amnestic syndromes (AAS) are transient and mostly reversible disorders of short-term episodic memory caused by structural and/or functional alterations of brain structures involved in memory processes (such as the hippocampus) [1]. Among these clinical entities, transient global amnesia (TGA) is one of the most common [2,3,4].

On the other hand, a relevant group of patients with acute amnesia presents a distinctive form of focal Temporal lobe epilepsy (TLE) defined as Transient Epileptic Amnesia (TEA) [5,6,7].

Indeed, TEA is often misdiagnosed at presentation [8,9] due to the clinical similarities among AAS [10] and the absence of clear interictal epileptiform abnormalities (IEAs) during standard EEG (stEEG) recording (only 53.3% of patients with TEA present clear IEAs at stEEG) [11,12].

In this regard, quantitative EEG (qEEG) analysis of background activity and, in particular, Power spectral Analysis (PSA) has brought new techniques of EEG signal feature extraction in many neurological disorders [13,14].

In patients with TLE, Pellegrino et al. reported an increase in slow activity (delta and theta power) over the epileptic region when compared with healthy control subjects [15].

In TGA patients, a prior qEEG study demonstrated a decrease in absolute theta power in fronto-temporal regions and a decrease in absolute beta power in the parietal lobe during the acute phase [16], while another work showed a marked reduction of absolute beta 1 and alpha power in the left temporo-parietal regions far from amnesia attack when compared with healthy controls [17]. More recently, our group demonstrated an increased beta power in the uncus and parahippocampal gyrus of the affected hemisphere of patients with TEA when compared to those affected by TGA [18].

Despite this wealth of data in current literature, there are no studies exploring the application scenarios of qEEG analysis for a possible diagnostic work-up in AAS.

The aim of the present study was to retrospectively evaluate the use of qEEG analysis for differentiating TEA from TGA. To reach this goal, we computed and compared Power Spectral Density (PSD) of a cohort of TEA and TGA patients to search for possible spectral findings that differentiate these two conditions. Secondly, we proposed a multi-feature score, the EPIlepsy AMNesia (EPIAMNE) score, incorporating qEEG features of our sample of study with clinical and stEEG information. Finally, we evaluated and compared the diagnostic accuracy of our new score, clinical anamnestic findings and stEEG to determine the best diagnostic tool for the differential diagnosis between TEA and TGA.

## 2. Materials and Methods

### 2.1. Recruitment

For this study, we retrospectively analyzed files and information of ninety-one patients who were referred to our second-level facility of Campus Bio-Medico Hospital of Rome from 2010 to 2018 after initial access to a first-level emergency department due to one or more episodes of transient amnesia.

Patients were excluded if: (i) psychiatric comorbidities, dementia and diagnosis of a secondary cause for AAS (transient ischemic attack, metabolic encephalopathy, intoxication, psychogenic fugue, dissociative disorders, hypoglycemia) were identified; (ii) they assumed concomitant therapy with neuroactive molecules which may confound the interpretation of EEG analysis. In total, we selected eighty-three patients, while eight patients were excluded. For more details on patient selection, see the appropriate section of the work of Lanzone et al. 2018 [11].

According to Zeman’s criteria (1998) [19], evidence of epilepsy was supported through any combination of IEAs (spikes, spike and wave complexes or sharp waves) on EEG (stEEG recording and/or long-term EEG monitoring), reports of classically epileptic features simultaneous to retrograde and/or anterograde amnesia, history and positive treatment response to antiseizure medications (ASMs) at follow-up. For the purpose of this study, we considered EEG to be abnormal if focal or diffuse interictal epileptiform transients (spike, sharp wave, or spike-and-wave discharges) or focal theta waves with spiky morphology were found.

Diagnosis of TGA was supported through the current clinical diagnostic criteria of Hodge and Warlow in 1990 [20]: (1) Presence of anterograde amnesia, which is witnessed by an observer; (2) no clouding of consciousness or loss of personal identity; (3) cognitive impairment limited to amnesia; (4) no focal neurological or epileptic signs; (5) no recent history of head trauma or seizures; (5) resolution of symptoms within 24 h; (6) mild vegetative symptoms (e.g., headache, nausea and dizziness) might be present during the acute phase. Among TGA patients, we selected a cohort with female/male ratio and median age similar to TEA population. Detailed study flow is shown in Figure 1.

### 2.2. stEEG Recording and Amnesis Collection

During the hospitalization in our neurology ward, all patients underwent stEEG monitoring 2–6 days after the acute onset of amnesia. At the time of the study our hospital was a second-level facility, so this delay in the acquisition of stEEG in our center was caused by the logistic time related to the transfer from the first-level emergency department to our Hospital.

At the time of recording, subjects were not symptomatic for amnesia, not disoriented in time and space, not confused or with other neurological signs. All EEGs were recorded during quiet wake with eyes closed with a 19-channel cap based on the 10–20 electrode placement system [21]. Signal was recorded with a 32-channel Micromed device (Systemì Plus software, version 1.05.0002; Micromed, Mogliano Veneto, Italy). All the EEGs were performed always using the same apparatus.

For EEG signal processing and filtering, impedance was kept below 5 KΩ, reference electrode was placed on the right mastoid bone, sampling rate was 256 Hz, A/D conversion was made at 16 bit and pre-amplifiers amplitude range was ±3200 µV. The following hardware filters were used: notch 50 Hz, low-pass 70 Hz and high-pass filters 0.3 Hz. EEGs with a constant artefactual activity that could not be removed by preprocessing and/or with more than one bad EEG channel (due to artefactual activity or high impedance) were excluded.

Detailed amnesic collection, neurological examination and brain MRI were collected. For the purpose of this work, we focused our attention on the presence (simultaneous or subsequent to amnesia) of manifestations suggestive of a possible origin of epileptic nature (including classical clinical signs of focal epilepsy reported in Zeman’s criteria): tremor, ataxia, spatial disorientation, olfactory hallucinations, staring, lip-shaking or other automatisms, aphasia and mild disorientation without consciousness alterations (MDwCA). We called this set of subjective symptoms and objective signs “symptoms atypical for TGA”.

Patients who presented EEGs with suspected epileptiform abnormalities or had normal stEEG but high clinical suspicion for focal epilepsy underwent a 16-channel 24 h long-term EEG monitoring (sampling rate 256 Hz, 0.3-Hz to 70-Hz band pass filter; Micromed Brain Spy System), which allowed us to record even nocturnal sleep in all subjects studied.

### 2.3. EEG Analysis

Offline data pre-processing of raw data was performed using the following pipeline: (1) 50-Hz notch filter; (2) bandpass filter between 0.3 and 70 Hz (linear phase finite impulse response filter) (3) visual inspection and subsequent manual rejection of IEAs and recording artifacts by a neurologist experienced in EEG and epilepsy; (4) correction for eye-blink, pulse and muscular activity using an automated Independent Component Analysis procedure [22]. Offline data pre-processing was performed using the Brainstorm Toolbox for Matlab [23].

We subsequently selected a total of 180 s of continuous epochs from the original resting state EEG free from relevant artifacts and epileptiform activities for further analysis. PSD over all 19 channels (Fp1, Fp2, F3, F4, C3, C4, P3, P4, F7, F8, T3, T4, T5, T6, O1, O2, Fz, Cz and Pz) and specifically for only temporal electrodes T3 and T4 was computed with the aim of obtaining a measure of global and regional cortical activity: we called these two power spectra Global PSD and Regional PSD, respectively. For both Global and Regional PSD, relative spectral power was calculated for the following power bands: delta (0.5–4 Hz), theta (4.5–7.5 Hz), alpha (8–12.5 Hz), beta (13–30 Hz) and gamma (30.5–60 Hz). Brainstorm Toolbox for Matlab and R studio were used for calculation and plotting of data, respectively.

### 2.4. Statistical Analysis and EPIAMNE Score Proposal

#### 2.4.1. Comparison of Regional and Global PSD

Data distribution was checked by Shapiro–Wilk’s test. We compared relative PSD of TEA and TGA patients for each frequency bands, for both Global and Regional PSD. Comparisons were performed using the statistical nonparametric Wilcox Signed Rank test. Alpha level was set at <0.05 for statistical significance. Alpha-inflation due to multiple comparisons was adjusted using the Holm-Bonferroni procedure; all the results are reported after correction.

We performed receiver operating characteristic (ROC) curve model on frequency bands which revealed to statistically differ between TEA and TGA. The ROC curve point showing the highest combination of sensitivity and specificity was selected as the optimum cut-off able to differentiate TEA from TGA patients. The optimal threshold of ROC curves was chosen on the base of the best Youden index using the function “coords” of the package pROC of R stat. R studio was used for the computation of data, while MedCalc software was used to plot ROC curves.

#### 2.4.2. EPIAMNE Score Proposal

On the basis of qEEG results and ROC curve analysis, we incorporated characteristics of our cohort in the multi-features score EPIAMNE, together with clinical and stEEG findings. For the details about EPIAMNE score construction see Section 3.5.

#### 2.4.3. Comparison of EPIAMNE Score with stEEG and Clinical Anamnestic Findings

ROC curve models were built to evaluate the diagnostic performance in differentiating TEA from TGA of EPIAMNE score, clinical anamnestic findings (that is the presence of symptoms atypical for TGA during the amnesic attack, pSymAT) and electroencephalographic abnormalities identification at stEEG recording. Specificity, sensitivity, area under the curve (AUC), positive predictive value (PPV) and negative predictive value (NPV) of each diagnostic tool were determined. R studio was used for statistical computation of data, while Matlab was used to plot ROC curve. Finally, we tested the statistical significance of the difference between the areas under ROC curves (AUC_EPIAMNE score_ vs. AUC_stEEG_, AUC_EPIAMNE score_ vs. AUC_pSymAT_, AUC_stEEG_ vs. AUC_pSymAT_) with De Longo et al. method using R studio.

## 3. Results

### 3.1. Patients’ Characteristics

Among the eighty-three selected patients with AAS, nineteen (22.89%) received a diagnosis of TEA. The mean age of TEA group was 67.3 years (first quartile = 64.5 years, third quartile = 74.5 years), ranging from 41 to 81 years. Females/males ratio was 14/5. Among TEA patients, twelve (63.16%) fulfilled all diagnostic criteria by Zenman et al. reported above. Seven patients (36.84%) were evaluated after the first episode of ictal amnesic attack (they fulfilled all Zeman’s criteria except seizures recurrence), of whom six (85.71%) achieved seizure-freedom after the introduction of ASM, while one patient (14.29%) was lost at follow-up. Demographic characteristics of our TEA cohort are reported in Table 1.

Sixty-four patients (77.11%) resulted as affected by TGA. For the purpose of the study, among TGA patients we selected a cohort of twenty-one subjects with a mean age of 64.7 years (first quartile = 60 years, third quartile = 71 years) ranging from 50 to 73 years, with a females/males ratio equal to 13/8. Demographic characteristics of our TGA cohort are shown in Table 2.

### 3.2. EEG Recordings

stEEG was abnormal in eight patients who received a diagnosis of TEA (42.1%). Fifteen patients (78.9%) underwent long-term EEG monitoring, and they all (100%) had clear IEAs; in six patients (31.6%), IEAs were only present during sleep. Information about EEG recording of our TEA cohort are shown in detail in Table 1.

In TGA population, eighteen patients (85.7%) had normal stEEG, while three patients (12.3%) presented mild EEG focal spiky transients during stEEG, although a diagnosis of focal epilepsy was excluded for not fulfilling other supportive Zeman’s criteria. Information about EEG recording of our TGA cohort are shown in detail in Table 2.

### 3.3. Clinical Anamnestic Findings

Compared to TGA, a significantly greater amount of TEA patients reported the occurrence of clear epileptic features in addition to anterograde/anterograde amnesia. Specifically, in four out of nineteen TEA patients (21.05%), amnesia was the only symptom reported, while the remaining subjects complained of the following neurological symptoms atypical for TGA: MDwCA (thirteen patients, 68.42%), language disturbance (two patients, 10.53%), “tremor” (one patient, 5.26%) and ataxia (one patient). Clinical characteristics of our TEA cohort are shown in Table 1.

Instead, only four TGA patients (19.05%) reported the occurrence of additional symptoms, in particular MDwCA (two patients, 9.52%), left hemianopsia (one patient, 4.76%) and nausea (one patient, 4.76%).

Clinical characteristics of our TGA cohort are shown in Table 2.

### 3.4. qEEG Features of TEA and TGA

We found a statistically significant difference between TEA and TGA in theta frequency band in Global PSD, with a higher relative theta power in TGA group (Bonferroni corrected *p*-value = 0.04). No significant differences were observed in the other frequency bands. The distribution of relative power in TEA and TGA for each frequency band for Global PSD is shown in Figure 2B.

ROC curve analysis of relative theta power performed on Global spectrum of TEA and TGA patients revealed an AUC equal to 0.75 (95% CI: 0.58, 0.89); with a threshold of 0.11 (ROC curve point showing the highest combination of diagnostic values), sensitivity was 90.48%, specificity was 57.89%, PPV 70.37%, NPV 84.62% and accuracy 75% (Youden index 0.48). Figure 3 depicts on ROC curve of global relative theta power.

No significant differences were observed when comparing TEA and TGA in Regional PSD. The distribution of relative power in TEA and TGA for each frequency band for Regional PSD is shown in Figure 2C.

### 3.5. The EPIAMNE Score

For the creation of our new proposed score, we highlighted the following features: (1) presence of electroencephalographic abnormalities at stEEG recording, (2) Theta relative power of Global PSD, (3) clinical characteristics of ictal amnesic attack apart from retrograde and anterograde amnesia. For each category, we assigned a dichotomous value (zero/one). In detail:We assigned one point to patients showing IEAs or focal theta waves with spiky morphology at stEEG recording, otherwise zero.We assigned one point to patients with a relative global theta power lower than the optimal cut-off value derived from ROC curves analysis of Global PSD, otherwise zero.Lastly, we assigned one point to patients who reported the pSymAT during the amnesia attack, otherwise zero. We did not include response to antiepileptic therapy and recurrence of amnestic episodes since they cannot be predicted or evaluated when observing the first attack.

For each patient, these values were then added to yield an omnibus score in the range 0–4. Figure 4 depicts on the detailed scoring system used for EPIAMNE score.

### 3.6. Diagnostic Accuracy of stEEG and Clinical Anamnestic Findings

ROC curve analysis of electroencephalographic abnormalities identification at stEEG in differentiating TEA from TGA revealed as follows: AUC was equal to 0.64, sensitivity was 42.1%, specificity 90.48%, PPV 72.73%, while NPV was 63.33% (Youden index 0.33). Instead, ROC curve analysis of the pSymAT revealed an AUC equal to 0.80, while sensitivity was 73.68%, specificity 85.71%, PPV 82.35% and NPV 78.26% (Youden index 0.59).

Diagnostic parameters of stEEG and clinical data are displayed in Figure 5.

### 3.7. Diagnostic Accuracy of the Epiamne Score

For each patient, we calculated the EPIAMNE score in the way described in the previous section. Median score of TEA patients was two points (first quartile = one point, third quartile = two points), ranging from one to three points; median score of TGA group was zero (first quartile = zero, third quartile = one point), ranging from zero to one point.

ROC curve analysis revealed as follows: AUC was equal to 0.94, while the optimal threshold with the highest combination of sensitivity and specificity was 1.5 (Youden index 0.68). Thus, patients with an EPIAMNE score ≥ 1.5 were classified as TEA, while patients with an EPIAMNE score < 1.5 were classified as affected by TGA; with this cut-off, sensitivity was 68.42% and specificity 100%, while PPV and NPV were equal to 100% and 77.78%, respectively.

Diagnostic parameters of EPIMANE score are displayed in Figure 5.

### 3.8. Comparison between the EPIAMNE Score, stEEG and Clinical Anamnestic Findings

Pairwise ROC curve comparison between electroencephalographic abnormalities identification at stEEG and our new proposed score revealed a *p*-value < 0.0001 (95 percent confidence interval: −0.43, −0.17; Z = −4.41), while the comparison between the pSymAT and our score demonstrated a *p*-value equal to 0.02 (95 percent confidence interval: −0.26, −0.03; Z = −2.43). Pairwise ROC curve comparison between pSymAT and electroencephalographic abnormalities identification at stEEG revealed a *p*-value of 0.16 (95 percent confidence interval: −0.38, 0.06; Z = −1.42).

## 4. Discussion

Our study is, to our knowledge, the first attempt to evaluate the use qEEG analysis in the differential diagnosis between TEA and TGA. We demonstrated that the use of qEEG analysis could improve the diagnostic accuracy of stEEG and clinical data (these results are discussed in Section 4.1 and Section 4.2). Secondary, we confirmed the existence of marked spectral differences between TEA and TGA (qEEG features of our cohort are discussed in Section 4.3).

### 4.1. EPIAMNE Score vs. stEEG

For the differential diagnosis of epilepsy, EEG is of great importance, but it’s a common experience that stEEG is of limited value in the detection of IEAs due to the brevity of the recording [24]. It is also well known that long-term EEG monitoring increases the diagnostic sensitivity of EEG [25,26] thanks to the duration of recording and the registration of NREM sleep phase [27], in which the amount of epileptiform abnormalities is proved to be increased both in generalized and focal epilepsy [28,29,30].

As in previous reports [9,31,32], our data confirmed that long-term EEG recording (particularly with prolonged studies including sleep) is able to identify abnormalities not seen in prior short-term EEG recording. However, the availability of long-term EEG monitoring is sometimes limited (even in the neurological ward) and wait time for prolonged outpatient ambulatory EEG is often long (weeks to months). Therefore, neurologists have limited instruments for the initial assessment of AAS, especially in the emergency clinical setting.

In this regard, our data provide evidence that an all-inclusive scoring system, like our EPIAMNE score, could contribute to advancing beyond the current state of the art by providing an instrument for differentiating TGA from TEA, particularly when short-term EEG recordings are unrevealing. Actually, EPIAMNE score was revealed to be more sensitive and with higher NPV than stEEG, thus emphasizing the improved ability of our score to reduce misdiagnosis of TEA compared to routine EEG recording. Moreover, ROC curve analysis and pairwise ROC curve comparison confirmed a statistically significant superiority of our proposed score in differentiating TEA from TGA when compared to electroencephalographic abnormalities identification at routine EEG.

### 4.2. EPIAMNE Score vs. Clinical Anamnestic Findings

From a clinical perspective, our results are partially consistent with TEA syndrome as previously described in detail in literature about AAS [1,8,9,33], confirming that, despite the presence of clinical similarities, TEA and TGA differ in many ways one from the other. Strangely, in our sample we observed a female predominance not consistent with epidemiological data previously reported [34]; this finding could be due to the low numerosity of the sample.

Being TEA part of TLE spectrum [9,35], it’s not surprising that a large percentage of TEA patients can experience the occurrence of further features that are strictly dependent on the underlying epileptic nature of TEA, such as typical ictal (e.g., language disturbance) and/or postictal manifestations (e.g., MDwCA, facial palsy and ataxia).

In our cohort, the presence of these ictal/postictal manifestations was superior in the prediction of TEA than electroencephalographic abnormalities identification at stEEG, thus emphasizing the importance of clinical presentation in the differential diagnosis between TEA and the nonepileptic TGA (being also a core component of the Zeman’s diagnostic criteria) [9,19]. More generally, epilepsy is primarily a clinical diagnosis [36], [37] and to confirm the crucial role of clinical manifestation, home-video recording is proving to be an effective method for an objective evaluation of transient clinical episodes not witnessed by medical personnel [38].

Regarding clinical presentation of our TGA cohort, some patients reported some atypical findings, that is hemianopsia and MDwCA. In the literature, TGA patients with atypical clinical presentation have been described (the so-called “TGA-plus syndrome”) [39]. Our small group of patients with “TGA-plus syndrome” has been confirmed to be affected by TGA, since at follow-up their EEGs were devoid of IEAs and they did not report other amnesia episodes.

Regarding radiological findings at brain MRI, only one TGA patient has a hippocampal punctate diffusion-weighted imaging (DWI) lesion at brain MRI in our cohort. This low rate of DWI+ lesion detection could be explained by the fact that our patient underwent brain MRI more than 4 days after the onset of symptoms, as for standard waiting time in our Institution. Actually, it’s well known that the detection rate of FLAIR and DWI lesions at brain MRI in TGA highly depends on the time interval from the acute phase (it decreases days after symptom onset) [40,41].

Taking into consideration the comparison with EPIAMNE score, our new proposed score proved to be significantly superior in differentiating TEA from TGA, as demonstrated by AUC and pairwise ROC curve comparison. Current clinical criteria for TGA do not allow clinicians to exclude other acute amnestic syndromes that occur in emergency situations [42]. Thanks to its high specificity and high PPV, EPIAMNE score could help clinicians to confirm the diagnosis of TGA, especially when further instrumental analysis, such as brain MRI and long-term EEG monitoring, are not immediately available in the emergency room.

### 4.3. qEEG Features of TEA and TGA

Our analysis revealed marked spectral differences in background activity between the two samples; in particular we demonstrated a significant increase in relative theta power in TGA patients in Global PSD.

Whatever the trigger of amnesia syndrome is, it’s well known that the hippocampus and more specifically the CA1 area [43,44,45] is the final common pathway to the abrupt onset of severe amnesia and the locus of brain modifications associated with TGA [46]. These discoveries have allowed researchers to establish a link between corticohippocampal circuit anomalies and the neurochemical mechanisms at the base of TGA [45,47,48]. In particular, hippocampus and corticohippocampal feedback loops regulate theta activity in the neocortex [49] so an increase in overall theta activity may reflect the dysfunction of inhibitory interneurons in the hippocampal CA1 field [50]. Both the amount of theta rhythm, specifically when affecting cortical regions that belonged to the posterior medial network [51], and the functional connectivity in the theta band [52] could explain episodic memory impairment during the acute stage of TGA.

Moreover, there is strong evidence of an enduring hippocampal dysfunction in TGA even after the complete clinical recovery [17,46] as demonstrated by neuropsychological investigations [53,54], Positron Emission Tomography studies [55] and qEEG studies [17]. Therefore, in this perspective, the increase in relative theta power over Global and Regional PSD in our study may reflect the persistence of corticohippocampal circuit malfunction with subsequent perturbation of overall brain networking. However, to be confirmed, these speculations would require more ad hoc designed studies in this regard.

Taking into consideration Regional spectrum of TEA and TGA patients, we did not find any statistically significant difference between TEA and TGA. Previous work already demonstrated the absence of significant interictal spectral differences between temporal regions of TEA and TGA, except for an asymmetric increase in Beta band power over the affected (where the epileptic focus is located) hemisphere of TEA patients when compared to TGA [18]. Previous qEEG work agrees on the existence of spectral asymmetry between affected and non-affected hemisphere in patients with TLE [15,18].

In our study, we did not focus on the presence of interhemispheric spectral differences, not being the description of spectral differences between TEA and TGA the primary aim of the present work. Indeed, regional PSD was calculated over both sides of the hemispheres, thus hiding possible interhemispheric spectral differences located in temporal lobes.

## 5. Limitations

This study is not free from limitations.

The first limitation regards the retrospective nature of the study design as collection was performed according to clinical reports and last follow-up visits of our patients in our Institution. This reduces the quality of data collection.

Taking into consideration our TEA cohort, it should be noted that we enrolled patients who reported amnesia attacks highly suggestive of focal seizures, despite not all patients satisfied Zeman’s core criteria of recurrence. Moreover, the numerosity of our sample is limited due to the rarity of TEA [56] and the cohort of study is heterogenous (patients do not share the same epileptic focus and some of them were discharged with the diagnosis of structural focal epilepsy).

Finally, our spectral results could not be representative of qEEG features of TEA and TGA during the acute phase since stEEGs were collected days after the complete resolution of the amnestic spells.

Although we demonstrated an improved diagnostic accuracy of EPIAMNE score for the differential diagnosis between TEA and TGA, we were not able to evaluate the effect that such an approach could have on the clinical management of patients with AAS (e.g., possible improvement of diagnostic and therapeutic delay). Future randomized prospective studies with a sufficiently large sample size are required to clarify whether the use of qEEG analysis could significantly impact on the management of patients with TEA, especially in the emergency setting where diagnostic instruments are limited.

## 6. Conclusions and Future Prospective

EPIAMNE score revealed to be not only more accurate in the differential diagnosis between TEA and TGA (especially when compared to routine EEG that is usually done in the emergency room), but also simple and immediate for interpretation, thus meeting the clinical need for a reliable and more accurate tool for the early diagnostic classification of AAS. Therefore, our score could give to neurologists an “added value” for differentiating TEA from TGA. It should also be noted that our analysis used conventional 19-channels EEG, which is inexpensive and widely available for clinical practice in most epilepsy centers.

To conclude, in our opinion qEEG analysis would eventually contribute to the development of a decision-making tool for the differential diagnosis of AAS, especially in the emergency setting. Larger studies are required to leverage more information to design differential diagnostic algorithms for AAS including qEEG analysis.

## Figures and Tables

**Figure 1 brainsci-12-01632-f001:**
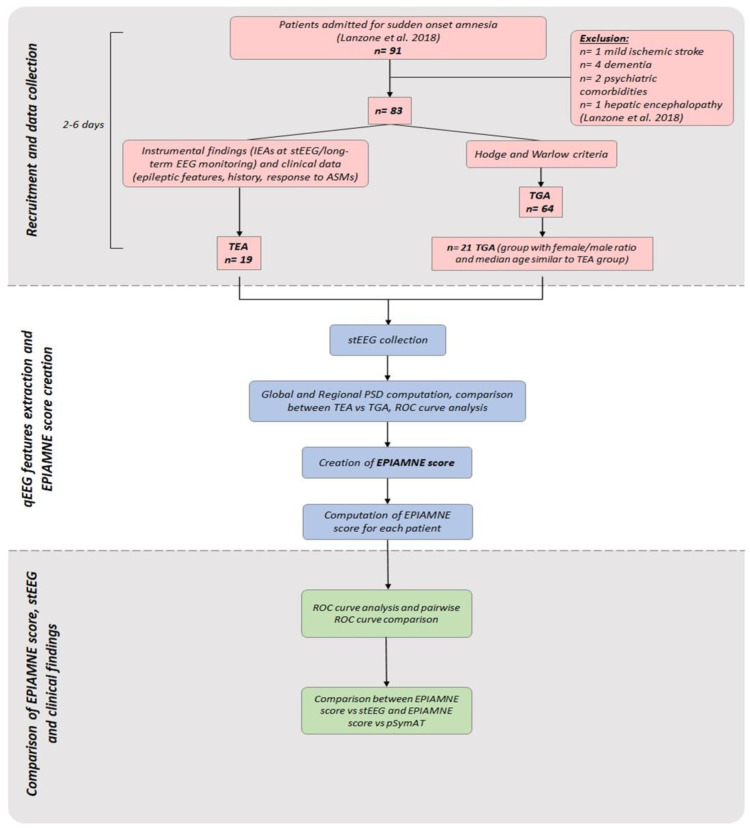
Flow chart of the study. ASMs (antiseizure medications), EPIAMNE score (EPIlepsy AMNEsia Score), IEAs (intercritic EEG anomalies), pSymAT (presence of symptoms atypical for TGA), stEEG (standard EEG), TEA (transient epileptic amnesia), TGA (transient global amnesia) [11].

**Figure 2 brainsci-12-01632-f002:**
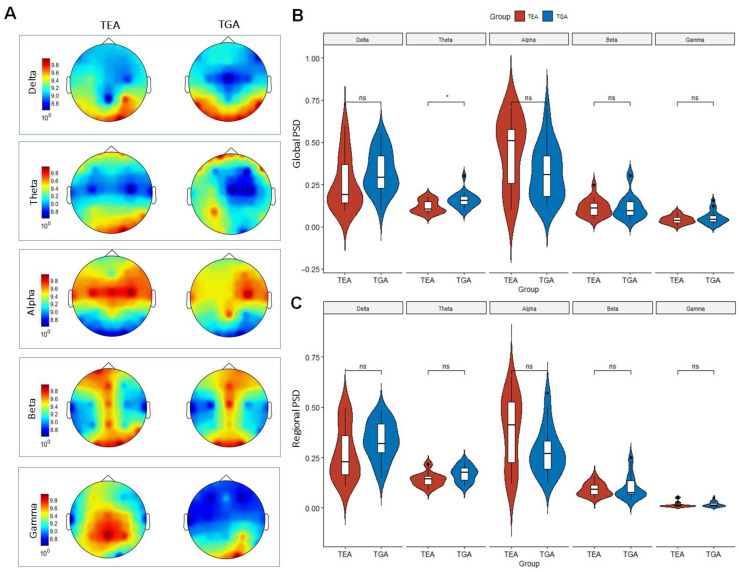
(**A**) Scalp distribution of PSD of TEA and TGA patients for each frequency band (Delta, Theta, Alpha, Beta and Gamma). (**B**) Violin plots show the distribution of relative power in TEA and TGA for each frequency band for Global PSD. (**C**) Violin plots show the distribution of relative power in TEA and TGA for each frequency band for Regional PSD. ns (not statistically significant), PSD (power spectral density), TEA (Transient Epileptic Amnesia), TGA (Transient Global Amnesia), * (*p* < 0.05).

**Figure 3 brainsci-12-01632-f003:**
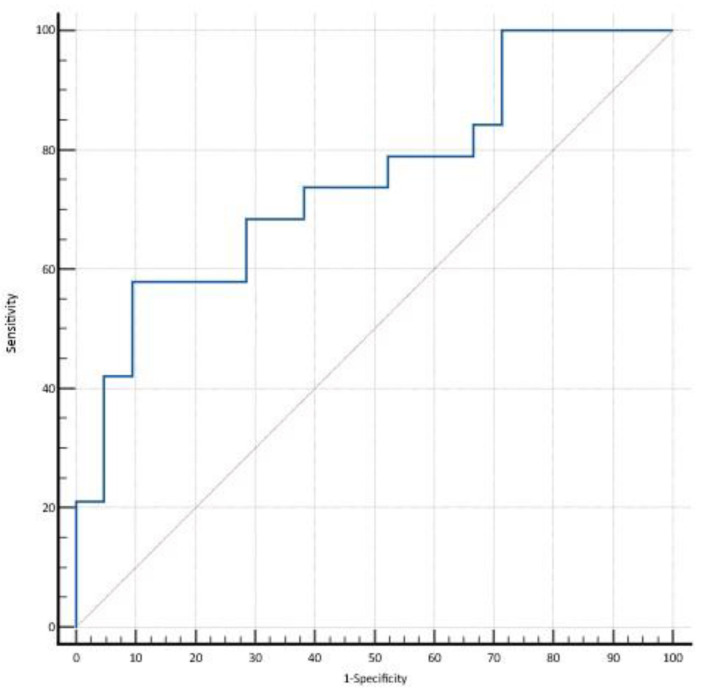
ROC curve analysis of theta relative power performed on Global PSD of TEA and TGA patients.

**Figure 4 brainsci-12-01632-f004:**
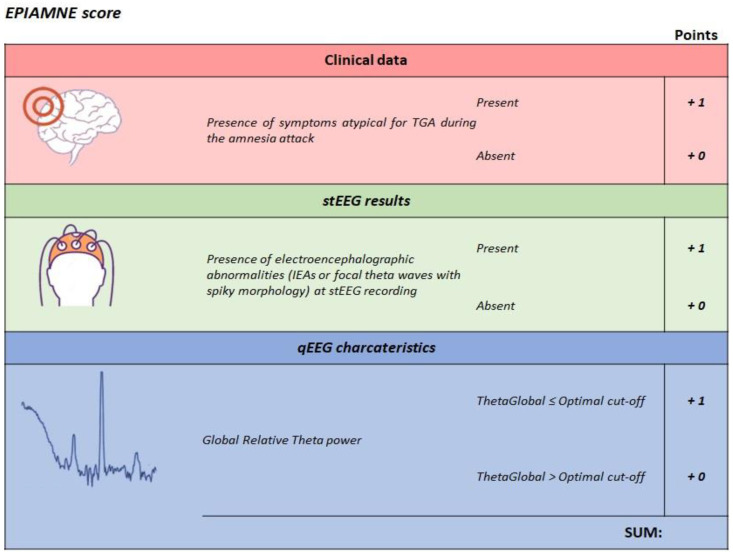
Construction of EPIMANE score. EPIAMNE score (EPIlepsy AMNEsia Score), IEA (interictal epileptic anomalies), qEEG (quantitative EEG), stEEG (standard EEG), TGA (transient global amnesia).

**Figure 5 brainsci-12-01632-f005:**
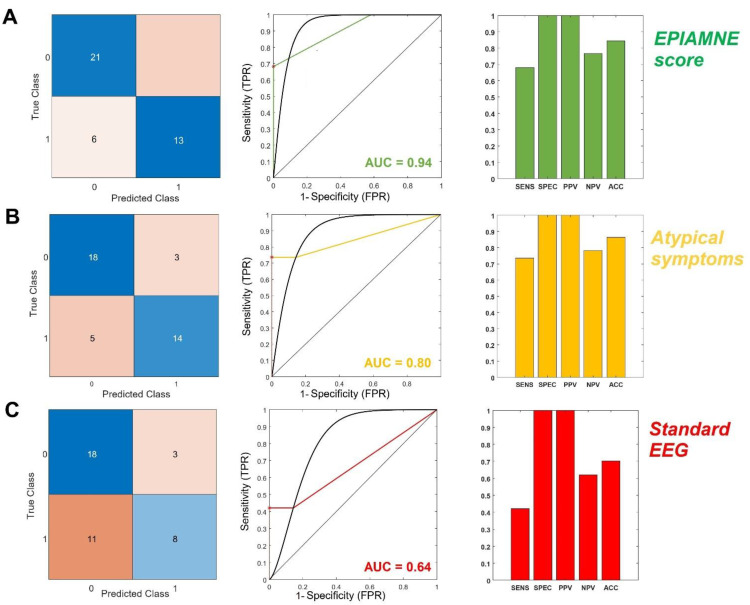
ROC curve comparison of the EPIAMNE score (**A**), pSymAT (atypical symptoms) (**B**) and standard EEG (**C**). ACC (accuracy), AUC (Area Under the Curve), EPIAMNE score (EPIlepsy AMNEsia Score), FPR (false positive rate), NPV (Negative Predictive Value), PPV (Positive Predictive Value), SENS (sensitivity), SPEC (specificity), TPR (true positive rate).

**Table 1 brainsci-12-01632-t001:** Clinical, demographic and EEG characteristics of our TEA cohort.

Patients	1	2	3	4	5	6	7	8	9	10	11	12	13	14	15	16	17	18	19
Age	41	69	75	70	75	73	63	67	66	61	76	68	48	69	81	60	74	75	67
Gender	M	F	F	M	M	F	F	M	F	F	F	F	F	F	F	M	F	F	F
stEEG	Pos	Neg	Neg	Pos	Neg	Neg	Pos	Neg	Neg	Neg	Neg	Neg	Neg	Pos	Pos	Pos	Pos	Neg	Pos
EEG 24 H	Pos	Pos	Pos	Pos	Pos	Pos	Pos	Pos	Pos	Pos	Pos	Pos	Pos	Neg	-	Pos	-	Pos	-
Focus	FT bil	FT bil	FT left	FT right	FT left	FT bil > left	FT left	FT bil	T left	FT left	FT left	T right	T right	T bil > right	FT right	FT left	T right	FT left	FT left
Duration	2	2,5 h	5 h	3 h	12 h	12 h	6 h	2 h	3 h	2 h	15 min	24 h	30 min	1 h	16 h	15 min	1 h	2 h	1 h
Recurrence	4	0	2	5	3	0	0	2	2	2	3	5	0	0	0	3	3	0	5
Other symptoms	Not reported	Mild confusion	Mild confusion, Language disturbance	Mild confusion	Mild confusion	Mild confusion	Headache	Mild confusion	Mild confusion	Mild confusion	Mild confusion	Mild confusion	Not reported	Not reported	Tremor, Ataxia	Not reported	Mild confusion	Mild confusion, Facial palsy	Mild confusion, Language disturbance
MRI	Neg	Neg	Neg	Neg	Neg	Neg	Neg	Temporal venous ectasia	Neg	Thalamic cavernoma	Neg	Cystic pinealoma	Neg	Neg	Hydrocephalus	Neg	Neg	Cerebellar cavernoma	Neg
Epilepsy	TLE	TLE	SFE	TLE	TLE	TLE	TLE	TLE	TLE	SFE	TLE	TLE	TLE	TLE	SFE	TLE	TLE	TLE	TLE
Therapy	*	CBZ	LVT	LVT	LVT	LVT	LTG	LVT	LVT	LVT	LVT	ZNS	LTG	LTG	LVT	LVT	LVT	LVT	LVT
Outcome	*	SF	SF	SF	SF	*	SF	SF	SF	*	SF	*	SF	SF	SF	SF	*	SF	Less than
																			1/y

Bil (bilateral), CBZ (carbamazepine), FT (frontotemporal), h (hours), LVT (levetiracetam), min (minutes), Neg (Negative), Pos (positive), TLE (temporal lobe epilepsy), SF (seizure-free), SFE(symptomatic focal epilepsy), T (temporal), LTG (lamotrigine), ZNS (zonisamide), * (lost at follow-up).

**Table 2 brainsci-12-01632-t002:** Clinical, demographic and EEG characteristics of our TGA cohort.

Patiets	1	2	3	4	5	6	7	8	9	10	11	12	13	14	15	16	17	18	19	20	21
Age	65	60	73	58	72	51	50	66	57	72	63	73	72	71	70	60	69	60	71	58	68
Gender	F	F	F	M	F	F	M	M	M	M	F	F	F	F	M	F	M	F	M	F	F
stEEG	Neg	Neg	Neg	Neg	Pos	Neg	Neg	Neg	Neg	Neg	Neg	Neg	Neg	Neg	Pos	Neg	Neg	Neg	Neg	Neg	Pos
Duration	6 h	3 h	3 h	3 h	1 h	3 h	4 h	4 h	4 h	5 h	8 h	2 h	4 h	4 h	4 h	13 h	12 h	13 h	3 h	1 h	7 h
Recurrence	0	2	0	0	3	0	0	4	2	0	0	2	0	3	0	0	2	0	0	0	0
Other symptoms	Not reported	Not reported	Not reported	Mild confusion	Not reported	Not reported	Left Hemianopsia	Not reported	Not reported	Not reported	Not reported	Mild confusion	Nausea and vomiting	Not reported	Not reported	Not reported	Not reported	Not reported	Not reported	Not reported	Not reported
MRI	Neg	Neg	Neg	Neg	Neg	Neg	Neg	Neg	Neg	Neg	Neg	Pos (DWI+)	Neg	Neg	Neg	Neg	Neg	Neg	Neg	Neg	Neg
Therapy	No	ASA	ASA	ASA	Plavix	ASA	ASA	ASA	ASA	ASA + Plavix	ASA	ASA	Plavix	Anticoagulant	No	No	ASA	No	ASA	No	No
														therapy							

ASA (acetylsalicylic acid).

## Data Availability

Raw unprocessed data supporting the findings of the present study are available from the corresponding author, upon reasonable request.

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
