# Peer review of "EPIAMNE: A New Scoring System for Differentiating Transient EPIleptic AMNEsia from Transient Global Amnesia"

_brainsci, 2022, doi:10.3390/brainsci12121632_

Round 1
Reviewer 1 Report
The purpose of this study is to describe a method to improve the accuracy of diagnosis between TEA and TGA.
The study proposes the calculation of a mixed clinical/qEEG score that is supposed to be superior to the purely clinical diagnosis in distinguishing TGA from TEA.
I will make several remarks on the clinical aspects of the study
1) the retrospective nature of the study reduces the quality of the data collection.
2) There are several atypical findings in the sample of TGA subjects. One subject has hemianopia and two have confusion, which sheds some doubt on the diagnosis. Clinically atypical TGAs have been described (see Piffer et al but in these cases most of the subjects have typical MRI-DWI hippocampal hyperintensities (e.g. Piffer S et al.. J Neurol Sci 441 (2022) 120349).
3) Only 1 TGA subject has MRI abnormalities which is below average. It would be crucial to know the timing of IRM, since MRI is positive for hyperintensities mostly in 12-24h post-onset.
3) the application of the EPIAMNE score results in 19% of TGA being classified as TEA which in my experience of TGA is below the clinical/MRI diagnosis
4) the advantage of the EPIAMNE score could be to miss fewer diagnoses of AET, but on the other hand the sample of AET presented by the authors verifies the ZEMAN criteria. It would be necessary to evaluate how the EPIAMNE score influences the management by following the patients prospectively.
Author Response
The purpose of this study is to describe a method to improve the accuracy of diagnosis between TEA and TGA. The study proposes the calculation of a mixed clinical/qEEG score that is supposed to be superior to the purely clinical diagnosis in distinguishing TGA from TEA. I will make several remarks on the clinical aspects of the study.
We thank the reviewer for the critical comments and useful suggestions that help us to improve the manuscript. We have taken into consideration all his/her comments and suggestions in the revised manuscript.
1) the retrospective nature of the study reduces the quality of the data collection.
We agree with the reviewer that the retrospective nature of our study is a limitation as data collection was performed according to clinical reports and last follow-up visits of our patients in our institution. We have acknowledged more clearly this notion in the “Limitations” section of the revised manuscript (page 16, lines 414-416).
2) There are several atypical findings in the sample of TGA subjects. One subject has hemianopia and two have confusion, which sheds some doubt on the diagnosis. Clinically atypical TGAs have been described (see Piffer et al but in these cases most of the subjects have typical MRI-DWI hippocampal hyperintensities (e.g. Piffer S et al.. J Neurol Sci 441 (2022) 120349).
We thank the reviewer for emphasizing this important point and for sharing this interesting work with us, it highlights the difficulty in differentiating acute amnestic syndromes. We included this work in the “Discussion” section of the revised manuscript (page 15, lines 356-361).
Our small group of patients with “TGA-plus syndrome” has been confirmed to be affected by TGA, since at follow-up their EEGs were devoid of interictal epileptiform abnormalities and since they did not report other amnesia episodes.
We revised the clinical history of our patients; they reported a symptom more compatible with “mild disorientation” without consciousness alterations rather than “confusion”. We modified this term in the revised manuscript (page 4, lines 120-121; page 6, Table 1; page 8, Table 2; page 9, line 218 and line 223; page 14, line 347).
3) Only 1 TGA subject has MRI abnormalities which is below average. It would be crucial to know the timing of IRM, since MRI is positive for hyperintensities mostly in 12-24h post-onset.
We agree with the reviewer. Indeed, for differentiating “TGA-plus syndrome” from other neurological causes of amnesia it will be paramount to perform brain MRI within 72 hours from symptoms onset. Unfortunately, all our TGA patients underwent brain MRI more than 4 days after the onset of symptoms, as for standard waiting time in our Institution. We have discussed more specifically this point in the “Discussion” section of the revised manuscript. (page 15, lines 363-369).
4) the application of the EPIAMNE score results in 19% of TGA being classified as TEA which in my experience of TGA is below the clinical/MRI diagnosis
We thank the reviewer for underlying this important issue and we apologize for the unclearness of this particular section.
According also to the suggestions of Reviewer 2, we performed a deep revision of the statistical analysis and of our results. Ultimately, we decided to exclude regional relative theta power from the computation of EPIAMNE score (page 11, lines 245-247 and lines 257-259) since our results showed only a trend corresponding to an increase in TGA but not significantly different after Bonferroni correction (p= 0.15).
The diagnostic parameters of the revised score are the following: sensitivity 68.42 %, specificity 100%, positive predictive value 100% and negative predictive value 77.78 % (page 13, lines 289-297; page 14, lines 302-305). Thus, our score was able to correctly classify each TGA patient of our cohort. We revised the “Discussion” section (page 14, lines 332-334; page 15, lines 371-377, line 382 and line 399) and the “Conclusion and future prospective” section (page 16, lines 430-431 and line 434) in the light of these new results. We discussed the possible explanations for the lack of significant differences in Regional PSD between TGA and TA in the “Discussion” section of the revised manuscript (pages 15-16, lines 402-412).
We agree that MRI signs can be of paramount importance in revealing mild radiological alterations that have been showed to be associated with the acute onset of TGA. However, it is important to highlight that the detection rate of FLAIR and DWI lesions in TGA highly depends on the time interval from the acute phase (it decreases days after symptom onset) (Sung-Hye You et al. in Neuroradiology of 2021; M. Scheel et al. in Clinical Neuroradiology of 2012) and several emergency departments are unable to perform brain MRI in the acute phase. We did not include MRI criteria in the construction of EPIAMNE score; it will be of interest to include radiological findings in a diagnostic tool for differentiating acute amnestic syndromes.
Regarding diagnostic accuracy of clinical diagnosis, EPIAMNE score revealed to be more accurate than clinical anamnestic findings alone in differentiating TEA from TGA. Moreover, current clinical criteria for TGA do not allow clinicians to exclude other acute amnestic syndromes usually observed in emergency setting (M. Sparaco et al. in Journal of Clinical Medicine of 2022). In literature, there are several papers showing that patients may present with all the classical features fitting TGA diagnosis, but only later they will show evidence for the presence of epilepsy (Kapur, Journal of Neurology, Neurosurgery, and Psychiatry 1993; Zemanet al., Neurol Neurosurg Psychiatry 1998; Hodges, WB Saunders 1991; Bilo et al., Epilepsia 2009). Thanks to its high specificity and high positive predictive value, EPIAMNE score could help clinicians to confirm the diagnosis of TGA, especially when further instrumental analysis, such as brain MRI and long-term EEG monitoring are not immediately available in the emergency room.
4) the advantage of the EPIAMNE score could be to miss fewer diagnoses of AET, but on the other hand the sample of AET presented by the authors verifies the ZEMAN criteria. It would be necessary to evaluate how the EPIAMNE score influences the management by following the patients prospectively.
We thank the reviewer, and we agree with his/her statement. Zeman criteria were the “gold standard” for the diagnosis of TEA in our study. However, to confirm the presence of interictal epileptiform abnormalities on EEG recordings (one of Zeman’s core criteria) our patients had to undergo long-term EEG monitoring, which is high cost and often unavailable in many first level epilepsy centers.
Even without the use of long-term EEG recording, our score had a high sensitivity in differentiating TEA from TGA when compared to standard EEG. Morover, it was revealed to be more accurate in differentiating TEA from TGA when compared to clinical anamnestic findings. Therefore, our score has the potential to be an useful tool in the early assessment of TEA and TGA, especially in those cases when further instrumental analysis (such as brain MRI and long-term EEG monitoring) is not immediately available in the emergency setting.
As suggested by reviewer, we acknowledged more clearly in the “Limitations” section of the revised manuscript that future prospective studies are necessary to evaluate how the EPIAMNE score would eventually improve the clinical management of TEA and TGA patients. (page 16, lines 427-433).

Reviewer 2 Report
General comments: A retrospective evaluation of the use of qEEG analysis for differentiating TEA from TGA. Are these techniques for qEEG widely available and, hence, the scoring (EPIAMNE) widely applicable? I suspect not, and that we will have to continue to rely on clinical diagnosis. Is there any evidence that clinical differential diagnosis of TGA vs TEA is often wrong?
Particular comments:
Line 78 and elsewhere: what is “fixation/anterograde amnesia”?
Lines 83-84: “the current clinical diagnostic criteria proposed by Caplan in 1985”. Caplan proposed boundaries within the diagnostic category of TGA, not criteria. The vast majority of writers on TGA do not used Caplan as the standard for TGA diagnosis.
Line 84: “modified by Hodge in 1990”! To read “modified by Hodges and Warlow in 1990”. These are the most frequently used criteria for TGA.
Figure 1: “Caplan and Hodge criteria”. Such do not exist.
Figure 1 legend: “EAs (intercritic EEG anomalies)” vs Line 37: “interictal epileptiform abnormalities (IEAs)”. Inconsistency.
Lines 97-98: “During the hospitalization in our neurology ward, all patients underwent stEEG monitoring 2-6 days after the acute onset of amnesia.”? TGA patients seldom require hospital admission, certainly not for 6 days!
Lines 114-116: “(tremor, ataxia, spatial disorientation, olfactory hallucinations, staring, lip-shaking or other automatisms, aphasia, confusion). We called this set of subjective symptoms and objective signs “symptoms atypical for TGA”.”. Is the implication that these are “symptoms typical for TEA”? Surely not, other than olfactory hallucinations, staring, lip-shaking or other automatisms?
Line 150: “ROC curve point showing the highest combination of sensitivity and specificity”. Vague: max Youden index? max Euclidean index? other?
Line 177: “Females/males ratio was 14/5.”. Odd, doesn’t TEA have a male predominance, about 3:1 (Baker et al., Brain Communications 2021;3:fcab038), hence the inverse of this group?
P9, lines 202-203: “information about EEG recording of our TGA cohort are shown in detail in Table 1.” To read “Table 2”.
Line 214: “hemianopasia” to read “hemianopia”?
Lines 215-216: “we used the term “confusion” to indicate severe attention impairment reported by patients and/or bystanders.”. Wouldn’t this contravene Hodges and Warlow TGA criteria?
Line 225: “Figure 1” to read “Figure 2”.
Line 237: “Figure 2” to read “Figure 3”.
Line 240: “Bonferroni corrected p= 0.15”. So not statistically significant (cf. preceding line)?
Line 261: “At least, we assigned” to read “Lastly, we assigned”?
Line 261: “pSymAT”. Has this abbreviation been previously defined?
Line 278: “Figure 3” to read “Figure 4”.
“Figure 3” middle panels: x axis is labelled “Specificity (FPR)”! FPR = 1 – specificity.
Line 287: “the highest combination of sensitivity and specificity was equal to 1.5 points”. Why present these data thus when previously you have specified sens, spec, PPV, NPV?
Line 296: “are displayed in Figure 4.” to read “are displayed in Figure 5.”
Line 308: not sure Figure 5 adds anything to what is shown in “Figure 3” (actually Figure 4).
Lines 311-312: “Our study is, to our knowledge, the first attempt to evaluate the use qEEG analysis in the differential diagnosis of AAS.”. Not so, the comparison has been restricted to TGA vs TEA, excluding other AAS.
Line 334: “more sensible” to read “more sensitive”? Ditto Lines 359, 415.
Line 355: “(Ricci et al. 2021)” to read “[38]”?
References passim: no pagination, except 20, 29, 46.
Ref 46: authors correct?
Author Response
General comments: A retrospective evaluation of the use of qEEG analysis for differentiating TEA from TGA. Are these techniques for qEEG widely available and, hence, the scoring (EPIAMNE) widely applicable? I suspect not, and that we will have to continue to rely on clinical diagnosis. Is there any evidence that clinical differential diagnosis of TGA vs TEA is often wrong?
We appreciate the reviewer’s opinion. Quantitative EEG features for EPIAMNE score construction were extracted from a conventional 19-channels EEG (widely available in every epilepsy center) using Brainstorm toolbox for MATLAB (F. Tadel et al., Comput Intell Neurosci, 2011), a free and widely available software. For this, we think that EPIAMNE score has the possibility to be widely accessible.
There are several works and case series in literature, which show that patients may present with all the classical features fitting TGA diagnosis, but only later they will show evidence for the presence of epilepsy (Kapur, Journal of Neurology, Neurosurgery, and Psychiatry 1993; Zemanet al., Neurol Neurosurg Psychiatry 1998; Hodges, WB Saunders 1991; Bilo et al., Epilepasi 2009). Our score turned out to be more accurate in differentiating TEA from TGA. Moreover, thanks to its high specificity and high positive predictive value, EPIAMNE score could help clinicians to confirm the diagnosis of TGA, especially when further instrumental analysis, such as brain MRI and long-term EEG monitoring are not immediately available in the emergency setting.
We agree that further prospective studies with a sufficiently large sample size are required to verify if EPIAMNE could significantly impact on the management of patients with TEA.
Particular comments:
Line 78 and elsewhere: what is “fixation/anterograde amnesia”?
We apologize for this oversight. We actually meant “retrograde and anterograde amnesia”. For the sake of clarity we edited it accordingly (page 2, line 78; page 11, line 253).
Lines 83-84: “the current clinical diagnostic criteria proposed by Caplan in 1985”. Caplan proposed boundaries within the diagnostic category of TGA, not criteria. The vast majority of writers on TGA do not used Caplan as the standard for TGA diagnosis.
We agree with the reviewer. We edited it in the revised manuscript (page2, line 84).
Line 84: “modified by Hodge in 1990”! To read “modified by Hodges and Warlow in 1990”. These are the most frequently used criteria for TGA.
We thank the reviewer for his/her suggestion. We edited it in the revised manuscript (page 2, line 84).
Figure 1: “Caplan and Hodge criteria”. Such do not exist.
We agree with the reviewer. We edited it in the revised manuscript (page 3, Figure 1).
Figure 1 legend: “EAs (intercritic EEG anomalies)” vs Line 37: “interictal epileptiform abnormalities (IEAs)”. Inconsistency.
Amended (page 3, Figure 1).
Lines 97-98: “During the hospitalization in our neurology ward, all patients underwent stEEG monitoring 2-6 days after the acute onset of amnesia.”? TGA patients seldom require hospital admission, certainly not for 6 days!
At the time of the study, our hospital was a second-level facility, so this delay in the acquisition of standard EEG in our center was due to the logistic time related to the transfer from the first-level emergency department to our Hospital. We have specified it more clearly in “Materials and Methods” section of the revised manuscript (page 4, lines 98-101).
Lines 114-116: “(tremor, ataxia, spatial disorientation, olfactory hallucinations, staring, lip-shaking or other automatisms, aphasia, confusion). We called this set of subjective symptoms and objective signs “symptoms atypical for TGA”.”. Is the implication that these are “symptoms typical for TEA”? Surely not, other than olfactory hallucinations, staring, lip-shaking or other automatisms?
What we called “symptoms atypical for TGA” are the symptoms and signs suggesting a possible origin of epileptic nature, including classical clinical signs of focal epilepsy reported in Zeman’s criteria. Anyway, no TEA patients in our population reported olfactory hallucinations, staring, lip-shaking or other automatisms. In particular, our patients complained symptoms such as mild disorientation without clear impairment of awareness, language disorders, facial palsy and ataxia more compatible with post-ictal rather than ictal symptoms. We have specified more clearly all these considerations in the “Material and Method” section of the revised manuscript (page 4, line 117-121).
Line 150: “ROC curve point showing the highest combination of sensitivity and specificity”. Vague: max Youden index? max Euclidean index? other?
We used the function “coords” of the package “pROC” of R stat which uses Youden Index to estimate the best threshold of ROC curve (RDocumentation- coords function). We have specified this information as suggested (page 5, lines 159-161). We also specified the Youden Index of the optimal threshold of Global relative theta power (page10, lines 241-242), stEEG (page 12, line 275), pSymAT (page 12, line 277) and EPIAMNE score (page 13, lines 293-294).
Line 177: “Females/males ratio was 14/5.”. Odd, doesn’t TEA have a male predominance, about 3:1 (Baker et al., Brain Communications 2021;3:fcab038), hence the inverse of this group?
We thank the reviewer for underling this issue. In our sample, we observed a female predominance not consistent with epidemiological data previously reported (Baker et al., 2021); this finding could be probably due to the small sample size. We have discussed this consideration in the “Limitations” section of the revised manuscript (page 14, lines 342-344).
P9, lines 202-203: “information about EEG recording of our TGA cohort are shown in detail in Table 1.” To read “Table 2”.
We edited it accordingly (page 9, line 212).
Line 214: “hemianopasia” to read “hemianopia”?
We edited it accordingly (page 9, line 223; page 8, Table 2).
Lines 215-216: “we used the term “confusion” to indicate severe attention impairment reported by patients and/or bystanders.”. Wouldn’t this contravene Hodges and Warlow TGA criteria?
We agree with the reviewer in this point. We examined again the clinical history of our patients; they reported a symptom more compatible with “mild disorientation” without consciousness alterations rather than “confusion”. We modified this term in the revised manuscript (page 4, lines 120-121; page 6, Table 1; page 8, Table 2; page 9, line 218 and line 223; page 14, line 347).
Line 225: “Figure 1” to read “Figure 2”.
We edited it accordingly (page 10, line 233).
Line 237: “Figure 2” to read “Figure 3”.
We edited it accordingly (page 11, line 244).
Line 240: “Bonferroni corrected p= 0.15”. So not statistically significant (cf. preceding line)?
We apologize for this issue, and we agree that this specific methodology of our work needs further clarification. We performed a deep revision of the statistical analysis and of our results. Ultimately, we decided to exclude regional relative theta power from the computation of EPIAMNE score (page 11, lines 245-247 and lines 257-259) since our results showed only a trend corresponding to an increase in TGA but not significantly different after Bonferroni correction (p= 0.15).
The diagnostic parameters of the score are the following: sensitivity 68.42 %, specificity 100%, positive predictive value 100% and negative predictive value 77.78 % (page 13, lines 289-297; page 14, lines 302-305). We revised the “Discussion” section (page 14, lines 332-334; page 15, lines 371-377, line 382 and line 399) and the “Conclusion and future prospective” section (page 16, lines 436-437 and lines 440-441) in the light of these new results. In “Discussion” section, we explained why we did not find any statistically significant difference in Regional PSD (pages 15-16, lines 402-412).
Line 261: “At least, we assigned” to read “Lastly, we assigned”?
We edited it accordingly (page 11, line 260)
Line 261: “pSymAT”. Has this abbreviation been previously defined?
We defined this acronym previously in page 5, line 170
Line 278: “Figure 3” to read “Figure 4”.
We edited it accordingly (page 13, line 283)
“Figure 3” middle panels: x axis is labelled “Specificity (FPR)”! FPR = 1 – specificity.
We edited it accordingly (page 13, Figure 4)
Line 287: “the highest combination of sensitivity and specificity was equal to 1.5 points”. Why present these data thus when previously you have specified sens, spec, PPV, NPV?
We agree with the reviewer that this part of the manuscript is confounding. For the sake of clarity, we simplified this section and we removed superfluous and conflicting elements. In the revised manuscript, we left the only cut-off value provided by ROC curve analysis (i.e. 1.5).
Line 296: “are displayed in Figure 4.” to read “are displayed in Figure 5.”
Amended.
Line 308: not sure Figure 5 adds anything to what is shown in “Figure 3” (actually Figure 4).
Amended.
Lines 311-312: “Our study is, to our knowledge, the first attempt to evaluate the use qEEG analysis in the differential diagnosis of AAS.”. Not so, the comparison has been restricted to TGA vs TEA, excluding other AAS.
We agree with the Reviewer and have now removed the term AAS from this sentence (page 14, line 310).
Line 334: “more sensible” to read “more sensitive”? Ditto Lines 359, 415.
We edited it accordingly (page 14, lines 332-333).
Line 355: “(Ricci et al. 2021)” to read “[38]”?
We edited it accordingly (page 15, line 356)
References passim: no pagination, except 20, 29, 46.
We edited it accordingly (pages 17-20)
Ref 46: authors correct?
We edited it accordingly (page 19, line 566)

Round 2
Reviewer 1 Report
I read the corrections proposed by the authors and I am OK with their propositions